# Transcriptional Regulation of Liver-Type OATP1B3 (Lt-OATP1B3) and Cancer-Type OATP1B3 (Ct-OATP1B3) Studied in Hepatocyte-Derived and Colon Cancer-Derived Cell Lines

**DOI:** 10.3390/pharmaceutics15030738

**Published:** 2023-02-23

**Authors:** Bastian Haberkorn, Dennis Löwen, Lukas Meier, Martin F. Fromm, Jörg König

**Affiliations:** Institute of Experimental and Clinical Pharmacology and Toxicology, Friedrich-Alexander-Universität Erlangen-Nürnberg (FAU), 91054 Erlangen, Germany

**Keywords:** Lt-OATP1B3, Ct-OATP1B3, *Lt-SLCO1B3*, *Ct-SLCO1B3*, colorectal carcinoma, ZKSCAN3, SOX9, HNF1α

## Abstract

Due to alternative splicing, the *SLCO1B3* gene encodes two protein variants; the hepatic uptake transporter liver-type OATP1B3 (Lt-OATP1B3) and the cancer-type OATP1B3 (Ct-OATP1B3) expressed in several cancerous tissues. There is limited information about the cell type-specific transcriptional regulation of both variants and about transcription factors regulating this differential expression. Therefore, we cloned DNA fragments from the promoter regions of the *Lt-SLCO1B3* and the *Ct-SLCO1B3* gene and investigated their luciferase activity in hepatocellular and colorectal cancer cell lines. Both promoters showed differences in their luciferase activity depending on the used cell lines. We identified the first 100 bp upstream of the transcriptional start site as the core promoter region of the *Ct-SLCO1B3* gene. In silico predicted binding sites for the transcription factors ZKSCAN3, SOX9 and HNF1α localized within these fragments were further analyzed. The mutagenesis of the ZKSCAN3 binding site reduced the luciferase activity of the *Ct-SLCO1B3* reporter gene construct in the colorectal cancer cell lines DLD1 and T84 to 29.9% and 14.3%, respectively. In contrast, using the liver-derived Hep3B cells, 71.6% residual activity could be measured. This indicates that the transcription factors ZKSCAN3 and SOX9 are important for the cell type-specific transcriptional regulation of the *Ct-SLCO1B3* gene.

## 1. Introduction

Transport proteins belonging to the superfamily of SLC (solute carrier) or ABC (ATP-binding cassette) transporters are expressed in almost all cells and tissues including pharmacologically important cell types such as enterocytes, kidney proximal tubule epithelial cells and hepatocytes [1,2,3,4]. In addition to their expression in healthy tissues, transport proteins are also expressed in multiple cancerous tissues, where they can mediate drug resistance [5,6,7,8].

One of these transport proteins, relevant for pharmacokinetics and effects of drugs, is the liver-type organic anion transporting polypeptide 1B3 (Lt-OATP1B3) [9]. Lt-OATP1B3 (gene symbol *Lt-SLCO1B3*) is a member of the SLCO/SLC21 transporter family [10]. OATP1B3 has been characterized as an uptake transporter localized in the basolateral membrane of human hepatocytes [9]. Substrates of the human OATP1B3 protein include endogenous substances as well as xenobiotics including several widely prescribed drugs [11,12,13]. In 2001, Abe and coworkers detected the expression of the OATP1B3 mRNA in several cancerous tissues and cancer-derived cells [14]. Further investigations by Nagai et al. demonstrated that the identified mRNA encodes a second OATP1B3 variant, now referred to as the cancer-type OATP1B3 protein (Ct-OATP1B3) [15]. This variant results from alternative splicing of the *SLCO1B3* gene. The *Ct-SLCO1B3* gene possesses a unique first exon (exon 1*) located in intron 3 of the *Lt-SLCO1B3* gene [15] (intronic numbering is based on the Gene bank entry NM_019844.4). The Ct-OATP1B3 protein encoded by this splice variant lacks the first 28 amino acids of the Lt-OATP1B3 protein [16]. Recently, we could demonstrate that in contrast to the localization of the Lt-OATP1B3 protein in the plasma membrane, the Ct-OATP1B3 protein is localized intracellularly in the membrane of lysosomes [17]. There, Ct-OATP1B3 is likely to mediate the transport of substances (e.g., the kinase inhibitor encorafenib) into lysosomes, reducing the cytoplasmic concentration of these drugs. Application of an OATP1B3 inhibitor led to a better survival of cancer cells revealing a new mechanism of drug resistance.

*Ct-SLCO1B3* mRNA has been detected in several cancerous tissues including breast, prostate, and colorectal cancer [18,19,20,21]. In normal and cancerous tissues, Lt- and Ct-OATP1B3 seem to be differentially regulated. Lee and coworkers [18] demonstrated immunostaining of OATP1B3 (without differentiation between the Lt- and the Ct-OATP1B3 variant) in 81% of colon adenocarcinomas, whereas no staining could be detected in normal colonic mucosa samples. Sun et al. [16] investigated the *Ct-SLCO1B3* mRNA expression in matched pairs of cancerous and normal tissue of 39 colon cancer patients and detected *Ct-SLCO1B3* mRNA in 87.2% of those colon cancer tissue samples, but only in 2.6% of the adjacent normal tissue samples. 

Data regarding the cell type-specific regulation of *Lt-* and *Ct-SLCO1B3* mRNA expression are limited. Nagai et al. demonstrated that *Ct-SLCO1B3* mRNA expression could be regulated by an alternative promoter located up to −553 bp upstream of the transcriptional start site of the *Ct-SLCO1B3* variant [15]. This potential promoter region showed markedly increased luciferase activity in human colon-derived LS180 cells and in pancreatic cancer-derived PK45p cells. These findings could be verified by Han and colleagues [22]. They demonstrated that the hypoxia-inducible factor 1α (HIF-1α) influences the expression of the *Ct-SLCO1B3* mRNA. Interestingly, a detailed comparison between the *Lt-* and *Ct-SLCO1B3* promoter regions has not been performed so far. 

To gain insights into the cell type-specific regulation of *Lt-* and *Ct-SLCO1B3* mRNA expression, we cloned several DNA fragments of varying length located in front of the transcriptional start sites of both variants into a reporter gene vector. We analyzed their luciferase activity in hepatocellular carcinoma cell lines and in colorectal cancer cell lines. The region around −100 bp upstream of the transcriptional start site of the *Ct-SLCO1B1* gene was identified as the core promoter region. Therefore, we analyzed this DNA fragment in more detail by electrophoretic mobility shift assays (EMSA) and mutagenesis of potential transcription factor binding sites.

## 2. Materials and Methods

### 2.1. Materials

The Dual-Luciferase^®^ Reporter Assay System, the pGL3 Basic vector, and the pRL-TK vector were purchased from Promega GmbH (Walldorf, Germany). The iScript™ cDNA Synthesis Kit was obtained from Bio-Rad Laboratories GmbH (München, Germany). Minimum essential medium, DMEM/F12 (1:1), RPMI Medium 1640, Dulbecco’s phosphate buffered saline, fetal bovine serum, penicillin streptomycin solution, 0.05%-trypsin-EDTA solution, L-glutamine, sodium–pyruvate, MEM Non-Essential Amino Acids Solution (100×), the TOPO™ TA Cloning™ Kit, with pCR™2.1-TOPO™, the Lipofectamine™ 2000 Transfection Reagent, the LightShift™ Chemiluminescent EMSA Kit, and the Pierce™ BCA Protein Assay Kit were obtained from Thermo Fisher Scientific (Dreieich, Germany). The Monarch^®^ Genomic DNA Purification Kit was purchased from New England Biolabs GmbH (Frankfurt, Germany). The MyFi™ DNA Polymerase was obtained from Meridian Bioscience Inc. (Cincinnati, OH, USA). The Light Cycler^®^ FastStart DNA Master^PLUS^ SYBR Green I Kit was obtained from Roche Diagnostics GmbH (Mannheim, Germany). The QuikChange Multi Site-Directed Mutagenesis Kit was obtained from Agilent Technologies Deutschland GmbH (Waldbronn, Germany). The 12-well and 96-well cell culture plates were obtained from Greiner Bio-One (Frickenhausen, Germany). Human colon total RNA was purchased from Takara Bio Europe SAS (Saint-German-en-Laye, France).

### 2.2. Cell Culture

Human embryonic kidney 293 (HEK293; RRID:CVCL_0045), Caco-2 (RRID:CVCL_0025), HepG2 (RRID:CVCL_0027) and Hep3B (RRID:CVCL_0326) cells were obtained from ATCC and incubated at 37 °C and 5% CO_2_. The cell culture medium for HEK293 cells was minimum essential medium, supplemented with 10% heat-inactivated fetal bovine serum, 100 U/mL penicillin and 100 µg/mL streptomycin. For HepG2 and Hep3B cells, the above-mentioned HEK medium was additionally supplemented with L-glutamine (final concentration 2 mM). The cell culture medium for Caco-2 cells was DMEM with 10% heat-inactivated fetal bovine serum, 100 U/mL penicillin and 100 µg/mL streptomycin, 2 mM L-glutamine, 1 mM sodium-pyruvate and Non-Essential Amino Acids Solution. DLD1 (RRID:CVCL_0248) and T84 cells (RRID:CVCL_0555) as well as mRNA samples from ten colorectal carcinoma cell lines (Caco-2, CL14, DLD1, HCT116, LoVo, RKO, SW480, SW620, SW948, T84) were kindly provided by PD Dr. Britzen-Laurent and Prof. Dr. Naschberger (Department of Surgery, University Hospital Erlangen, Germany). DLD1, T84 and HEK293 cells have been authenticated using STR profiling within the last three years. For DLD1 cells, RPMI 1640 medium was used, and for T84 cells, DMEM/F12 (1:1) medium was used, both supplemented with 10% heat-inactivated fetal bovine serum, 100 U/mL penicillin and 100 µg/mL streptomycin. All experiments were performed with *mycoplasma*-free cells. Subcultivation was performed twice a week using trypsin 0.05%-EDTA 0.02% solution.

### 2.3. RNA Isolation and Quantitative RT-Polymerase Chain Reaction

Total RNA isolation and the quantitative RT-PCR (qRT-PCR) was performed according to [17]. In brief, 1 × 10^7^ cells were seeded on 10 cm cell culture dishes and the total RNA was isolated using the NucleoSpin RNA Plus Kit (MACHEREY-NAGEL GmbH & Co. KG; Düren, Germany). Per sample, 1 µg of total RNA was used for the cDNA synthesis using the iScript™ cDNA Synthesis Kit. Subsequently, the Light Cycler^®^ FastStart DNA Master^PLUS^ SYBR Green I Kit was used for the qRT-PCR. The expression of each gene was calculated via linear regression and normalized to the expression of the housekeeping gene *β-actin*.

### 2.4. MatInspector Analysis

For the prediction of potential transcription factor binding sites, the MatInspector tool from the Genomatix web page was used. The first 500 bp upstream of the transcriptional start site of the *Ct-SLCO1B3* variant served as template to calculate the matrix similarity between template and transcription factor binding sites.

### 2.5. Generation of the Reporter Gene Constructs and Conduction of the Luciferase Assays

The genomic DNA of HEK293 cells was isolated using the Monarch Genomic DNA Purification Kit (New England Biolabs; Frankfurt, Germany) according to the protocol. The genomic DNA of HEK293 cells was used because sequencing revealed almost 100% identity for the first 1000 bp in front of the transcriptional start sites with the database sequence of the upstream regions of the *Ct-* and *Lt-SLCO1B3* gene (Appendix A). The 2000 bp upstream region of the transcriptional start sites of *Ct-* and *Lt-SLCO1B3* were amplified with the MyFi^TM^ DNA Polymerase using 200 ng genomic DNA as template. The primers used for the cloning of the reporter gene constructs are listed in Table 1. Both fragments were subcloned into the pCR 2.1-TOPO TA-vector, sequenced and subsequently cloned into the luciferase reporter gene vector pGL3-Basic. The shorter reporter gene fragments were amplified using these 2000-Ct or 2000-Lt plasmids as templates and cloned into the pGL3-Basic vector as described above. Whereas a few bp exchanges could be detected between bp −1000 and −2000 of both fragments, the 100-Lt and 100-Ct DNA fragments investigated in detail in this study were 100% identical to the sequence of the database entry. DNA sequence analysis is shown in Appendix A.

A quantity of 3.5 × 10^5^ cells per well of the respective cell line was seeded in a 12-well plate. After 24 h, the cells were transiently transfected via lipofection with 1.5 µg of the respective pGL3-Basic firefly reporter gene plasmids and with 40 ng of the pRL-TK plasmid, serving as transfection control. The cells were incubated for 48 h and afterwards the medium was aspired, and the wells were washed with 800 µL purified water. To measure the luciferase activity, the Dual-Luciferase^®^ Reporter Assay System (Promega) was used. The cells were lysed with 200 µL of the diluted Passive Lysis Buffer (kit included) and incubated for 20 min on a plate shaker. Subsequently, 20 µl per well were transferred into a 96-well plate and put into the Centro LB960 Luminometer (Berthold Technologies GmbH & Co. KG, Bad Wildbad, Germany). There, 50 µL of the Luciferase Assay Reagent II were added and the firefly luciferase activity was measured for 12 s. Afterwards, 50 µL Stop and Glo Reagent were added and the renilla luciferase activity was measured for 12 s. 

To calculate the luciferase activity of the reporter gene constructs, the renilla luciferase intensity was first adjusted to the renilla intensity of the renilla plasmid cotransfected with the empty pGL3-Basic vector, resulting in the so-called ‘renilla factor’. This factor was multiplied by the firefly intensity of the respective sample and divided by the average firefly intensity of the empty pGL3-Basic vector. The luciferase activity is given as % of the empty pGL3-Basic vector.

### 2.6. Site-Directed Mutagenesis

Mutations into the potential binding sites of the 100-Ct reporter gene construct were introduced using the QuikChange Multi Site-Directed Mutagenesis Kit according to the protocol. The mutagenesis primers (Table 2) were designed according to the QuikChange Primer Design Tools from the Agilent web page. A total of 100 ng of the unmutated reporter gene vectors served as template. The successful mutation was validated via DNA sequencing.

### 2.7. Nuclear Protein Isolation

The nuclear proteins were isolated according to Łanuszewska et al. [23]. For each cell line, the cells of three 75 cm^2^ cell culture flasks were trypsinized and pooled in a 15 mL falcon. The pellet was washed with ice-cold PBS, resuspended in 3 mL resuspension buffer, and incubated on ice for 10 min. Afterwards, 18 µL Triton X-100 were added and mixed by vortexing. The samples were centrifuged for 15 min at 4 °C and 3000× *g*. The supernatant was removed, and the pellet was resuspended in a 1:1 mixture of low-salt buffer and high-salt buffer (total volume 250 µL). After incubating on ice, the samples were centrifuged for 30 min at 4 °C and 16,000× *g*. The supernatant was transferred to a 1.5 mL Eppendorf cup and aliquoted in 0.5 mL Eppendorf cups. These samples were snap-frozen with liquid nitrogen and stored at −80 °C. The protein concentration was measured with the Pierce^TM^ BCA Protein Assay Kit.

### 2.8. Electrophoretic Mobility Shift Assay

To analyze the interaction of the nuclear protein fractions of the different cell lines with the 100-Ct fragment, biotinylated probes (consisting of 60 bp long forward- and reverse-primers) were designed, covering bp −40 to bp −100 of the *Ct-SLCO1B3* upstream region which include all the investigated potential transcription factor binding sites. These primers were annealed to enable the interaction with the transcription factors. The unbiotinylated 60 bp DNA fragment served as competitor. To analyze the influence of the introduced mutations to the binding affinity of the transcription factors, a biotinylated DNA probe harboring all binding site mutations was also designed. The EMSA was performed using the LightShift™ Chemiluminescent EMSA Kit. Per sample, 70 fmol of biotinylated DNA probe was used and, if necessary, mixed with 20 µg nuclear protein. A total of 400 fmol of the unbiotinylated DNA served as competitor.

### 2.9. Statistical Analysis

The data are expressed as means ± SEM. The graphs and statistical analysis were conducted by using GraphPad Prism (RRID:SCR_002798; Version 5.01, 2007, GraphPad Software, San Diego, CA, USA). The statistical significance was analyzed using either One-way ANOVA with Bonferroni adjusted post hoc tests or a two-tailed unpaired Student’s *t*-test.

## 3. Results

### 3.1. Cloning of the CT- and Lt-SLCO1B3 Reporter Gene Constructs

Two potential promoter regions (Figure 1A) upstream of the transcriptional start site of the *Lt-SLCO1B3* gene and of the *Ct-SLCO1B3* splice variant were cloned into the pGL3-Basic reporter gene vector and designated as 2000-Lt (*Lt-SLCO1B3* promoter fragment) and 2000-Ct (*Ct-SLCO1B3* promoter fragment (Figure 1B)). Gene bank entries NM_019844.4 and NM_001349920.2 served as reference sequences for the *Lt-SLCO1B3* and *Ct-SLCO1B3* DNA fragments, respectively. The intronic sequences were analyzed using the Ensembl.org database entry SLCO1B3-202 (ENST00000381545.8). Both DNA fragments were gradually shortened leading to the different *Lt-SLCO1B3* and *Ct-SLCO1B3* reporter gene constructs (Figure 1B,C), all cloned into the reporter gene plasmid pGL3-Basic and used for the subsequent luciferase assays. Designation of the plasmids and the final length of the DNA fragments are summarized in Figure 1C. 

Next, expression of the *Ct-SLCO1B3* mRNA in the different colorectal carcinoma cell lines was analyzed by qRT-PCR (Figure 1D). Based on this analysis, Caco-2 (low *Ct-SLCO1B3* mRNA expression), DLD1 and T84 (high *Ct-SLCO1B3* mRNA expression) cells were used for further reporter gene analyses together with the hepatocellular carcinoma cell lines HepG2 and Hep3B.

### 3.2. Measurement of the Luciferase Activity in the Different Cell Lines

The luciferase activity of the different *Lt-* and *Ct-SLCO1B3* reporter gene constructs was analyzed in the hepatocellular carcinoma cell lines HepG2 (Figure 2A) and Hep3B (Figure 2B) and in the colorectal carcinoma cell lines DLD1 (Figure 2C), T84 (Figure 2D) and Caco-2 (Figure 2E). In Hep3B cells (Figure 2B), most *Lt-SLCO1B3* promoter constructs (except the 100-Lt and the 500-Lt fragments) showed significantly higher luciferase activity compared to the *Ct-SLCO1B3* reporter gene constructs. The luciferase activity of the 200-Lt construct measured in HepG2 cells was also significantly higher compared to the 200-Ct construct (Figure 2A).

However, in all colorectal carcinoma cell lines (Figure 2C–E), the *Ct-SLCO1B3* promoter constructs showed a significantly higher luciferase activity in most of the comparisons. In line with the qRT-PCR analysis of *Ct-SLCO1B3* mRNA expression, the luciferase activity of most reporter gene constructs was lower in Caco-2 cells compared to the luciferase activity in both other colorectal cancer cell lines. This demonstrates that based on the origin of the used cell lines, the expression of the OATP1B3 variants was differentially regulated.

Furthermore, in all three colorectal carcinoma cell lines, the 100-Ct promoter fragment showed the highest luciferase activity (Figure 2C–E), demonstrating that the core promoter of the *Ct-SLCO1B3* gene is located within this DNA fragment. Especially in DLD1 and T84 cells, the 100-Ct fragment showed a 5.5-fold and 25.5-fold higher luciferase activity, respectively, compared to the activity of the same fragment measured in Hep3B cells (Figure 3B). Therefore, the 100-Ct DNA fragment was analyzed in more detail.

### 3.3. Analysis of the 100-Ct Promoter Construct

To gain insights into the core promoter region of the *Ct-SLCO1B3* variant, binding of nuclear proteins to the 100-Ct DNA fragment was analyzed by an electrophoretic mobility shift assay (EMSA). A biotinylated DNA fragment covering the region between −40 bp and −100 bp upstream of the transcriptional start site (Figure 3A) was used as a DNA probe. For further experiments, nuclear extracts from Hep3B cells showing the lowest luciferase activity in the 100-Ct reporter gene analysis together with DLD1 and T84 cells, both showing the highest luciferase activity of the 100-Ct construct (Figure 3B), were used. The co-incubation of the biotinylated DNA probe with the nuclear proteins of all cell lines resulted in a strong band shift compared to the DNA fragment alone. This demonstrates binding of nuclear proteins to this specific DNA region. The addition of an excess of unbiotinylated DNA fragment resulted in a reduction in the band intensity of the shifted band showing a direct and inhibitable interaction of this DNA fragment with nuclear proteins (Figure 3C).

### 3.4. Mutational Analysis of Transcription Factor Binding Sites in the 100-Ct DNA Fragment

Potential transcription factor binding sites within the 100-Ct DNA fragment were analyzed using the MatInspector tool. From the in silico predicted binding sites within this region, the binding sites of the Zinc finger with KRAB and SCAN domains 3 (ZKSCAN3) transcription factor, the SRY-Box transcription factor 9 (SOX9) binding site and a binding site for hepatocyte nuclear factor 1α (HNF1α), a known transcription factor regulating *Lt-SLCO1B3* gene expression [24], were selected for mutational analysis because they showed a high matrix similarity in the MatInspector tool (Figure 4A). To abolish potential binding of the respective transcription factor, mutations were introduced within the respective transcription factor binding sites (Figure 4B). In addition, a triple mutated fragment (Triple-Mut) containing all three mutations was constructed (Figure 4B). The impact of all mutations related to the activity of the unmutated 100-Ct reporter gene fragment is shown in Figure 4C. 

In Hep3B cells, a slight reduction could be detected for the mutated SOX9 binding site, whereas the mutation of the ZKSCAN3 binding site and the triple-mutated DNA fragment significantly reduced the reporter gene activity. In DLD1 and T84 cells, the mutation of the SOX9 and of the ZKSCAN3 binding sites as well as the triple-mutated fragment significantly reduced the luciferase activity. In contrast, the mutation of the HNF1α binding site had no effect in T84 cells and a stimulatory effect in DLD1 cells.

The mutation-caused reduction in the luciferase activity of the mutated ZKSCAN3 binding site in the two colorectal carcinoma cell lines was significantly higher compared to the reduction in the Hep3B cell line (Figure 4D). T84 cells showed the lowest residual luciferase activity of 14.3 ± 0.8% followed by the activity measured in the DLD1 cells (29.9 ± 1.3%). In contrast, in Hep3B cells, the residual activity was 71.6 ± 2.3%. This is in line with the effect of the triple-mutated DNA fragment (Figure 4D). This differential influence of the introduced mutations could also be seen in an EMSA using the triple-mutated DNA fragment as DNA probe (Figure 4E). No difference could be detected between the wild-type DNA probe and the mutated DNA probe in Hep3B cells. However, the binding of nuclear proteins to the mutated DNA probe was reduced in both CRC cell lines. Interestingly, as demonstrated by qRT-PCR, the expression of the *ZKSCAN3* mRNA was significantly higher in DLD1 cells compared to the expression in normal colon tissue (Figure 4F) suggesting that this transcription factor is important for the regulation of the *Ct-SLCO1B3* mRNA expression.

## 4. Discussion

The aim of this study was to gain insights into the regulation of *Ct-SLCO1B3* mRNA expression. Ct-OATP1B3 (encoded by the *Ct-SLCO1B3* gene) is a splice variant of the drug transporter OATP1B3. Whereas the primary variant, designated in this manuscript as Lt-OATP1B3 (liver-type OATP1B3), is expressed in healthy human hepatocytes, the Ct-OATP1B3 (cancer-type OATP1B3) variant is expressed mostly in tumorous tissues such as colorectal or pancreatic carcinoma [16,20,25]. Little is known about the influence of transcription factors on the regulation of the mRNA expression of both variants. Therefore, we cloned several DNA fragments of the promoter regions of the *Lt-SLCO1B3* and *Ct-SLCO1B3* variants and analyzed their luciferase activity using hepatocellular and colorectal cancer cell lines.

Splicing events are important for the regulation of gene expression and alternative splicing of genes has a considerable impact on tumorigenesis [26]. Approximately 50% of all genes have distinct promoter regions differentially regulated in normal or tumorous tissues [27,28]. The Ct-OATP1B3 protein results from alternative splicing of the *SLCO1B3* gene at which the *Ct-SLCO1B3* gene possesses a first exon (designated as exon 1*), located in intron three of the *Lt-SLCO1B3* gene [15]. Recently, we could demonstrate that in contrast to the plasma membranous localization of the Lt-OATP1B3 protein, the Ct-OATP1B3 variant is localized in the lysosomal membrane mediating the transport of substances from the intracellular compartment into these vesicles [17].

Former studies investigating the regulation of the *Ct-SLCO1B3* mRNA expression demonstrated that the region between bp −553 to bp +4 upstream of the transcriptional start site of the *Ct-SLCO1B3* gene had a higher luciferase activity compared to the empty reporter gene vector [15]. This finding was verified by Han and coworkers investigating the promoter region between bp −1853 and bp +151 [22]. In the present study, we confirmed these results showing that in colon-derived DLD1, T84 and Caco-2 cells, all *Ct-SLCO1B3* reporter gene constructs (except the −200 bp construct) had a higher luciferase activity compared to the constructs from the *Lt-SLCO1B3* promoter region. Furthermore, we could demonstrate that the 100-Ct reporter gene construct had the highest luciferase activity in all colorectal cancer cell lines. In contrast, no difference between the luciferase activity of the 100-Lt and the 100-Ct fragment could be detected in Hep3B cells. This demonstrates that this region comprises the core promoter of *Ct-SLCO1B3* gene. 

Interestingly, in the hepatocyte-derived cell lines HepG2 and Hep3B, most *Lt-SLCO1B3* reporter gene constructs had comparable or, in the case of Hep3B cells, significantly higher luciferase activity compared to the *Ct-SLCO1B3* constructs. This reflects the hepatocellular background of both cell lines and the expression of the Lt-OATP1B3 protein in mostly healthy tissues [29], but contrasts with the expression of the *Ct-SLCO1B3* mRNA in tumorous tissues. However, this *Lt-SLCO1B3* reporter gene activity is lower, as expected. During the carcinogenesis of hepatocytes, several genes including the transport protein encoding genes are downregulated [30]. Especially uptake transporters of the SLCO/SLC21 family, including human OATP1B1 and OATP1B3, are downregulated in, e.g., the hepatocellular carcinoma cell line HepG2 [3,31]. This is probably due to the downregulation of stimulating transcription factors such as HNF1α and HNF4 [24,32]. Furthermore, it has been demonstrated that in 60% of investigated hepatocellular carcinoma samples, the *Lt-SLCO1B3* mRNA and the respective Lt-OATP1B3 protein are downregulated. The authors explained this with the overexpression of the hepatocyte nuclear factor 3β (HNF3β) and the transcriptional silencing of the *Lt-SLCO1B3* gene expression [30]. This may be one reason for the relatively low reporter gene activity of the *Lt-SLCO1B3* constructs in the hepatocellular carcinoma cell lines. Nevertheless, we have shown that in Hep3B cells, all *Lt-SLCO1B3* reporter gene constructs (except the 100-Lt and the 500-Lt construct; Figure 2) had a significantly higher activity compared to the *Ct-SLCO1B3* constructs, demonstrating that, based on the origin of the cell line, both OATP1B3 variants were differentially regulated. 

Besides the analysis of the different regulation of *Lt-* and *Ct-SLCO1B3* gene expression, the focus of this work was the analysis of the *Ct-SLCO1B3* promoter constructs in colorectal carcinoma cell lines. So far, only DNA methylation [33,34] and the transcription factor HIF1α [22,35] are known as potential regulators of the *Ct-SLCO1B3* mRNA expression. 

As shown in the luciferase assays using the reporter gene constructs (Figure 2), the 100-Ct fragment had the highest luciferase activity in all investigated CRC cell lines. This demonstrates that this region may comprise the core promoter of the *Ct-SLCO1B3* gene. Therefore, this DNA fragment was further analyzed for potential transcription factor binding sites using the MatInspector online tool. This matrices-based software package calculates the matrix similarity (M_sim_) between template and matrix. Total consensus is shown by a value of 1 [36]. In the 100-Ct fragment, three transcription factor binding sites for the transcription factors ZKSCAN3, SOX9 and HNF1α could be identified with M_sim_ values of 1, 0.94, and 0.89, respectively. Interestingly, ZKSCAN3 is known to be involved in the progression of colorectal carcinoma [37,38]. This is in line with our qRT-PCR data investigating the *ZKSCAN3* mRNA expression in normal colon total RNA and in the cancer-derived cell lines (Figure 4F). Whereas only a low expression of *ZKSCAN3* mRNA could be detected in normal colon total RNA, expression was slightly (not statistically significantly) increased in Hep3B and T84 cells and significantly increased in DLD1 cells. 

Furthermore, the transcription factor SOX9 is overexpressed in colorectal carcinoma, suggesting that SOX9 is also involved in tumorigenesis [39,40]. The role of both transcription factors for *Ct-SLCO1B3* mRNA expression could be confirmed by our mutagenesis analysis. Mutation of the ZKSCAN3 and SOX9 bindings sites of the 100-Ct fragment significantly reduced the luciferase activity measured in DLD1 and T84 cells (Figure 4C). Interestingly, mutation of both binding sites studied in Hep3B cells also reduced luciferase activity, suggesting that in hepatocyte-derived carcinoma cell lines, both transcription factors also influence gene expression, as long as the binding site is present in the regulatory region of the gene. Furthermore, the EMSAs using the triple-mutated biotinylated DNA fragment as DNA probe showed a reduced interaction of the nuclear protein fraction in comparison to the wild-type DNA fragment demonstrating a direct interaction between possible transcription factors with these mutated binding sites. However, the mutation of the transcription factor binding site of the known *Lt-SLCO1B3* transcription factor HNF1α [24] had no effect in T84 cells and a stimulatory effect in DLD1 cells. This indicates that HNF1α has no influence on the transcriptional regulation of *Ct-SLCO1B3* but regulates the expression of the *Lt-SLCO1B3* mRNA in the healthy human liver [24]. 

Taken together, we have analyzed the differential regulation of *Lt-* and *Ct-SLCO1B3* gene expression in cancer-derived cell lines of different origin and could demonstrate cell type-specific regulation. Additionally, we characterized the core promoter region of the *Ct-SLCO1B3* gene and identified the transcription factors ZKSCAN3 and SOX9 as potential regulators for its mRNA expression. These results could explain the tumor-specific expression of the *Ct-SLCO1B3* mRNA and the *Ct*-OATP1B3 protein. Furthermore, correlations between the expression of these transcription factors and the expression of *Ct-SLCO1B3* could help to further investigate the impact of Ct-OATP1B3 on the clinical outcome of cancer patients.

## Figures and Tables

**Figure 1 pharmaceutics-15-00738-f001:**
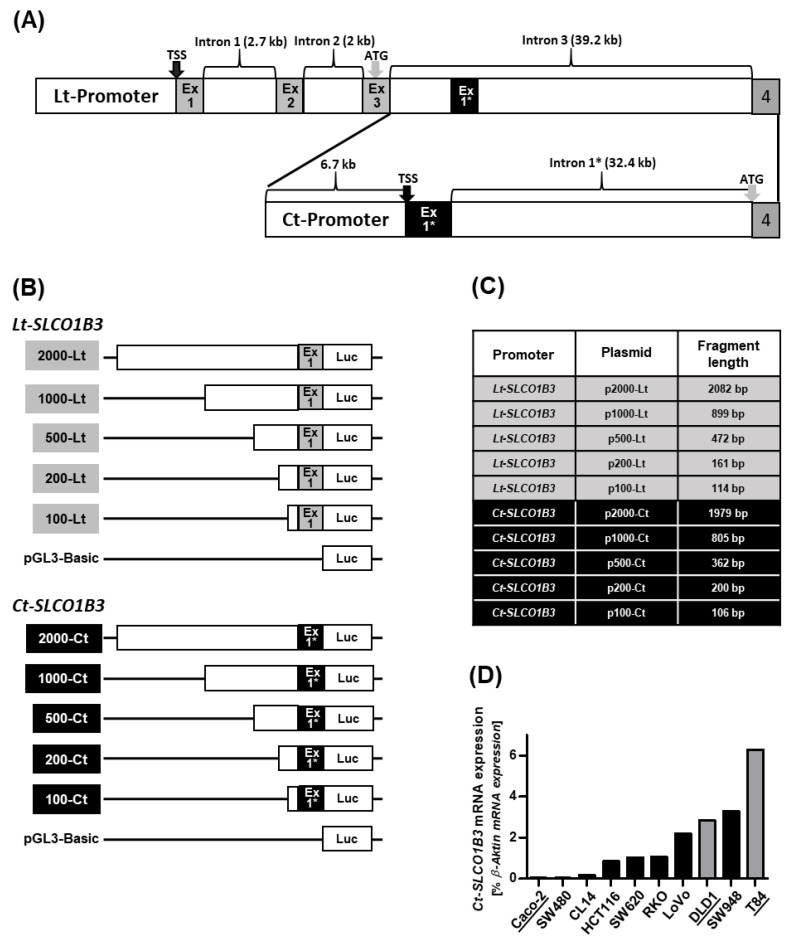
(**A**) Schematic overview of the upstream promoter regions of the *Lt-* and *Ct-SLCO1B3* genes. (**B**) For each promoter region, several reporter gene constructs were cloned with varying length (between 2000 bp and 100 bp) of the respective upstream region of the *Lt-* and *Ct-SLCO1B3* gene. The fragment lengths shown in (**C**) indicate the final length of the cloned fragments (without the respective exon overlap). (**D**) qRT-PCR analysis of *Ct-SLCO1B3* mRNA expression in ten colorectal carcinoma cell lines. Underlined names and grey boxes mark three cell lines used for further reporter gene analysis. TSS = transcriptional start site; ATG indicates the respective start codon; Ex1, Ex2, Ex3 = *Lt-SLCO1B3* exclusive exons; Ex1* = the alternatively spliced first exon of *Ct-SLCO1B3*; Intron 1* = first intron of the *Ct-SLCO1B3* gene; LUC = *firefly luciferase* gene within the pGL3-Basic vector.

**Figure 2 pharmaceutics-15-00738-f002:**
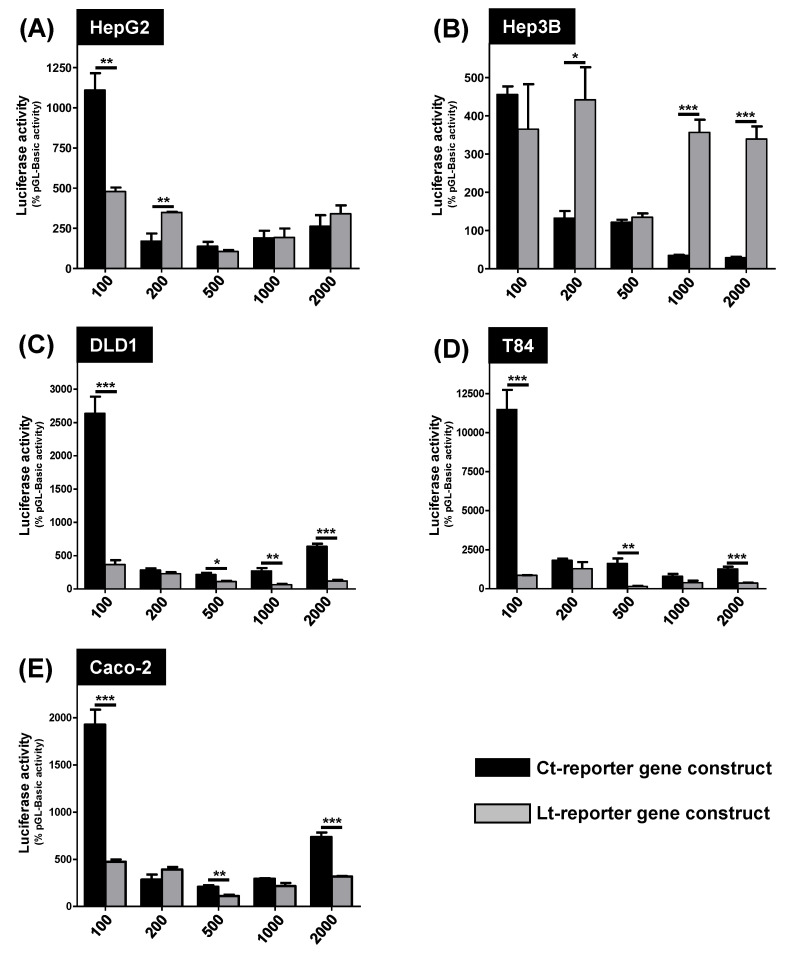
Comparison of the luciferase activities between *Lt-* and *Ct-SLCO1B3* reporter gene constructs in HepG2 (**A**), Hep3B (**B**), DLD1 (**C**), T84 (**D**) and Caco-2 (**E**) cells (n = 4). In each cell line, the reporter gene constructs with the same length were compared (Ct-constructs = black bars, Lt-constructs = grey bars). The luciferase activity is shown in % of the empty pGL3-Basic activity. * *p* < 0.05 Ct-reporter gene construct vs. Lt-reporter gene construct; ** *p* < 0.01 Ct-reporter gene construct vs. Lt-reporter gene construct; *** *p* < 0.001 Ct-reporter gene construct vs. Lt-reporter gene construct.

**Figure 3 pharmaceutics-15-00738-f003:**
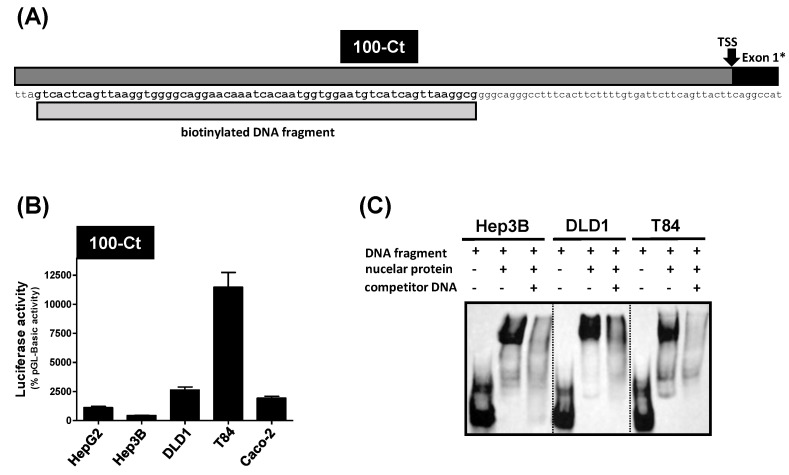
(**A**) Scheme of the 100-Ct reporter gene fragment. The light grey bar indicates the biotinylated DNA fragment which served as DNA probe for the EMSA shown in (**C**). (**B**) Comparison of the 100-Ct luciferase activities in the five tested cell lines (n = 4). (**C**) EMSA using the nuclear protein fraction of Hep3B, DLD1 and T84 cells and the biotinylated DNA fragment. The identical unbiotinylated fragment served as competitor DNA. TSS = transcriptional start site; Exon 1* = the alternatively spliced first exon of *Ct-SLCO1B3*.

**Figure 4 pharmaceutics-15-00738-f004:**
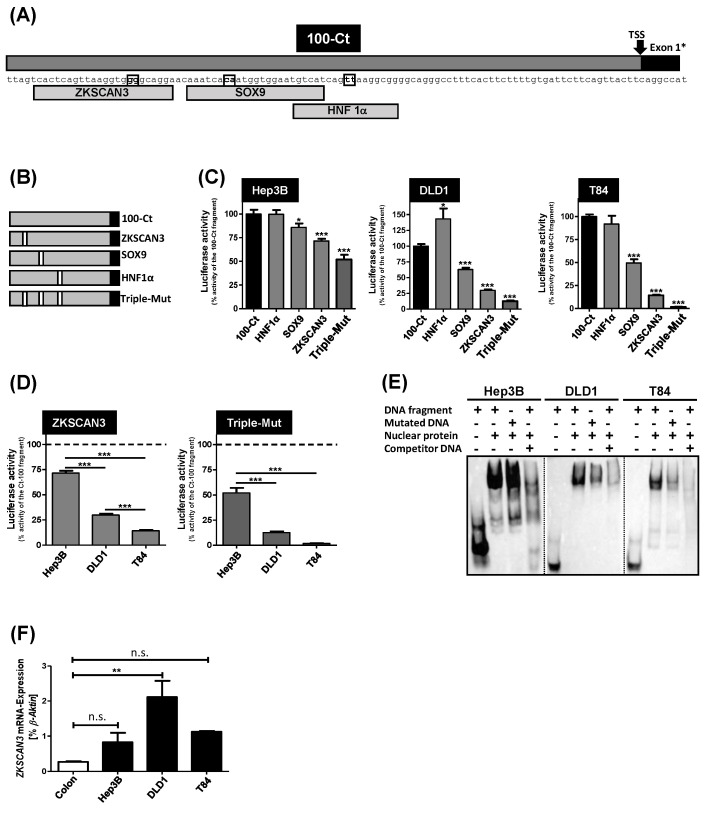
(**A**) Localization of potential transcription factor binding sites of the transcription factors ZKSCAN3, SOX9 and HNF1α within the 100-Ct fragment. The base pairs marked in the black boxes were mutated in the different constructs shown in (**B**). (**C**) Comparison of the luciferase activity of the mutated 100-Ct constructs in Hep3B, DLD1 and T84 cells with the activity of the non-mutated 100-Ct reporter gene fragment set to 100% (n = 8; Triple-Mut n = 4). (**D**) Influence of the ZKSCAN3 (n = 8) and Triple-Mut (n = 4) mutation on the luciferase activity of the three cell lines. (**E**) EMSA comparing the binding affinity of the nuclear protein fractions of Hep3B, DLD1 and T84 cells with the non-mutated biotinylated DNA fragment (second lane of each cell line) and the triple-mutated biotinylated DNA-fragment (third lane). The unbiotinylated DNA fragment served as competitor DNA (fourth lane). (**F**) Analysis of the *ZKSCAN3* mRNA expression in Hep3B, DLD1 and T84 cells and in normal colon total RNA (n = 3). * *p* < 0.05; ** *p* < 0.01; *** *p* < 0.001; n.s. = not significant. TSS = transcriptional start site; Exon 1* = the alternatively spliced first exon of *Ct-SLCO1B3*.

**Table 1 pharmaceutics-15-00738-t001:** Primers used for the cloning of the respective reporter gene constructs.

Primer	Sequence
2000-Ct.for	5′-tggaggcaaggaattgcaact-3′
1000-Ct.for	5′-gaaggaccaaggcaggcatc-3′
500-Ct.for	5′-ttcatgtgtgtccatgtgaagag-3′
200-Ct.for	5′-cagtcaaagggggttgttctct-3′
100-Ct.for	5′-ttagtcactcagttaaggtgggg-3′
Ct-Upstream.rev	5′-taactgaccattcccttacctgc-3′
2000-Lt.for	5′-gctgtcaagtagcagagacattgg-3′
1000-Lt.for	5′-acaactgtcctgtcagtgataagg-3′
500-Lt.for	5′-ccatgtgagatatccagtgtccatg-3′
200-Lt.for	5′-gataggcttctggggtgaactcc-3′
100-Lt.for	5′-ctgtttgcctaggacaatgacct-3′
Lt-Upstream.rev	5′-gcaactgcaacaagtccatcctt-3′

**Table 2 pharmaceutics-15-00738-t002:** Mutagenesis primers.

Primer	Sequence
oHNF1α-Mut	5′-ggtggaatgtcatcagccaaggcggggcaggg-3′
oSOX9-Mut	5′-aggtggggcaggaacaaatcaacatggtggaatgtcatcag-3′
oZKSCAN3-mut	5′-gttagtcactcagttaaggtgaagcaggaacaaatcacaatggt-3′

## Data Availability

The data presented in this study are available on request from the corresponding author.

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
