# Peer review of "Transcriptional Regulation of Liver-Type OATP1B3 (Lt-OATP1B3) and Cancer-Type OATP1B3 (Ct-OATP1B3) Studied in Hepatocyte-Derived and Colon Cancer-Derived Cell Lines"

_pharmaceutics, 2023, doi:10.3390/pharmaceutics15030738_

Round 1

Reviewer 1 Report

The authors studied the regulation of Lt-2 OATP1B3 and Ct-OATP1B3 at transcriptional levels in three cell lines derived from hepatocyte and colon cancer cells. OATP1B family is associated with multidrug resistance (MDR).

Comments:

1.        Why the authors used three cell lines derived from different cell types? Suppose they should use all cell lines derived from hepatocellular carcinoma (HCC), and use the cell lines derived from liver cells as controls!

2.        The English writing and expressing is very poor to be understood, and some words were used wrongly. Such as “Abstract: “So far, little is known about the cell-type specific regulation of Lt- or Ct-SLCO1B3 gene expression”, “To gain insights into this differ-ential regulation we cloned DNA fragments from both promoters and investigated their luciferase activity in hepatocellular and colorectal cancer cell lines”, “Both promoters were differentially regulated depending on the origin of the cell line used for reporter gene analysis”, and others.

3.        ABSTRACT: “We identified the first 100 bp upstream of the transcriptional start site as the core promoter region of Ct-SLCO1B3 gene expression and analyzed the in silico predicted binding sites for the transcription factors ZKSCAN3, SOX9 and HNF1α by mutational analysis”, Is it “gene expression”? So, how did you understand “transcription”?!

4.        ABSTRACT: “Whereas mutagenesis of the ZKSCAN3 binding site considerably reduced the transcriptional activity of Ct-SLCO1B3 in colorectal cancer cell lines (to 29.9 20 % in DLD1, 14.3 % in T84 cells), in liver-derived Hep3B cells 71.6 % residual activity could be measured”. It is poor for English expression!

5.        ABSTRACT: These results demonstrated that both SLCO1B3 gene variants are cell-type specifically regulated and that the transcription factors ZKSCAN3 and SOX9 are important for the transcriptional regulation of Ct-SLCO1B3.

The comments above are for ABSTRACTS only!

6.        Section 2.5 and 2.6 should be combined as one section!

7.        Section 3.1 Cloning of the reporter gene constructs: The primers for these cloning steps should be listed as a table! Also the DNA bands for their final products should be provided.

8.        The quality of Fig. 3-C is poor for publication!

9.        Every title of RESULTS was expressed unprofessionally! It should be a sentence with a verb!

10.    The manuscript is lack of a series of cellular results to validate the functions of this regulation!

Reviewer 2 Report

The work entitled „Transcriptional regulation of liver-type OATP1B3 (Lt-OATP1B3) and cancer-type OATP1B3 (Ct-OATP1B3) studied in hepatocyte-derived and colon-cancer-derived cell lines”  analyzes the regulation of transport proteins in normal cells and in cancer-derived cell lines. The study is very detailed and focused on the expression of transporter genes and the identification of the transcription factors. The above analysis is carried out on several cell lines, which is important and confirms the obtained results. In my opinion, there is no description of the meaning of the results in medicine and clinic. There is no information on how these studies can be used in practice.

Reviewer 3 Report

The article by Bastian Haberkorn et al presented the work on the topic “Transcriptional regulation of liver-type OATP1B3 (Lt- 2 OATP1B3) and cancer-type OATP1B3 (Ct-OATP1B3) studied in 3 hepatocyte-derived and colon-cancer-derived cell lines”. Theentirety of the work that has been done to this point is highly intriguing. In molecular biology and genetics, transcription factors (TFs) are proteins that selectively bind to certain regions of DNA. As a result, they regulate the process by which genetic information is copied from DNA to messenger RNA. Once they have attached themselves to DNA, these proteins can either encourage or inhibit the recruitment of RNA polymerase to particular genes, hence increasing or decreasing gene activity. When controlling how genes are expressed, transcription factors are absolutely necessary. Even though all of the cells in the body have the same DNA, those cells might have quite distinct functions depending on the present transcription factors. They attach themselves to certain DNA sequences inside the genes they are responsible for regulating; transcription factors attach themselves to one or more sequence locations known as transcription factor binding sites (TFBSs).

Overall, the research is well-executed and presented. I feel the following suggestions will aid in enhancing the quality of the study piece as a whole.

The initial impression of the studies quality might be enhanced by restructuring the abstract as it is not much reader-friendly.

On what hypothesis these (ZKSCAN3, 18 SOX9 and HNF1α) factors were employed in the study.

Considering that articles are always used as references and as the basis for further experimental setup, would you like to insert RNA isolation data in the methodology part before the qRT PCR section?

Would you mind explaining why this MatInspector analysis, compared to other existing techniques for identifying transcription factor binding sites, is more appropriate for your study?

Describe the key reasons for your decision to use the luciferase assay for transcription estimation.

 “HEK293”. Not discussed anything about it.

The language used in the manuscript in some places needs a major revision to conform to standard technical English.

The discussion needs to be better structured and framed. Grammatical and punctuation mistakes need to be explicitly corrected in this section. Perform a careful and comprehensive check of this.

Please add the relevant research article instead of referring to the review article wherever it is required as it takes credit from the actual scientific addition.

Describe in brief prospective findings from your studies in the molecular biology of transcription control, particularly as they relate to cancer biology and any further future direction on it.

The report indicates that the plagiarism percentage is 27%; you should strive to minimize this up to 15%.

Line 33. “where some of them play an” . Correct pronoun.

Line 43. “now referred as the cancer-type”. Add punctuation and correct preposition.

Line49. “protein [13], Recently,”. Two independent clauses correctly joined by the comma, consider correcting it.

Line 54. “drug”. Change it to plural noun.

Line 60. “whereas no staining could”. Add punctuation.

Line 65. “investigated”. The wrong verb tense is used, consider correcting it.

Line 79. “as core promoter region” Add article.

Line 139: “reporter gene vector pGL3- Basic resulting in the plasmids 2,000-Ct.pGL3 and 2,000-Lt.pGL3.” Consider this sentence for grammatical error.

Line 178. “After 30 min incubation on ice the samples were 178 centrifuged for 30 min at 4 °C and 16,000 x g.” Consider this sentence and add proper punctuation for increasing its meaning and readability.

Round 2

Reviewer 1 Report

I reviewed the manuscript two weeks ago. The authors revised their manuscript, especially revised their poor English writing.

But they still remain some comments of mine without response.

Such as:

1.        Why the authors used three cell lines derived from different cell types? Suppose they should use all cell lines derived from hepatocellular carcinoma (HCC), and use the cell lines derived from liver cells as controls!

2.        Section 3.1 Cloning of the reporter gene constructs: The primers for these cloning steps should be listed as a table! Also the DNA bands for their final products should be provided.

3.        Every title of RESULTS was expressed unprofessionally! It should be a sentence with a verb!

4.        The manuscript is lack of a series of cellular results to validate the functions of this regulation!

Author Response

Reviewer 1

General comment:

I reviewed the manuscript two weeks ago. The authors revised their manuscript, especially revised their poor English writing.

But they still remain some comments of mine without response.

Response

We are surprised about this comment because in the initial response letter, we have carefully addressed all points and suggestions raised by this reviewer including the four points mentioned below. Maybe something happened during the electronic submission process.

Comment 1:

Why the authors used three cell lines derived from different cell types? Suppose they should use all cell lines derived from hepatocellular carcinoma (HCC), and use the cell lines derived from liver cells as controls!

Response:

The Ct-OATP1B3 protein is not expressed in human hepatocytes but highly expressed in colorectal carcinoma tissue and cell lines derived from this cancer type. In contrast, the Lt-OATP1B3 protein is highly expressed in human hepatocytes and not expressed in cell lines originating from tumorous tissues. Therefore, cell lines originating from both tissues have to be used to study the cell-type specific regulation of both transporters. Using only cell lines derived from hepatocellular carcinoma or from liver cells would not be sufficient for studying the expression of the Ct-OATP1B3 variant because these cells will not express transcription factors necessary for regulating the Ct-OATP1B3 mRNA expression. For this reason, cell lines derived from different cell types were used.

Comment 2:

Section 3.1 Cloning of the reporter gene constructs: The primers for these cloning steps should be listed as a table! Also the DNA bands for their final products should be provided.

Response:

We have addressed this point in the revised version of our manuscript. Primers used for cloning of the genomic DNA fragments are listed in Table 1 and primers used for mutagenesis are listed in Table 2, both tables are located in the Materials and Methods section. As mentioned in the first response to this comment, cloning of the DNA fragments was a stepwise process and therefore, a combined DNA gel analysis for the amplification products is not available. However, as mentioned in the manuscript, all DNA fragments were sequenced after cloning to verify the correctness of the DNA fragments. If necessary, we can provide this sequence analysis.

Comment 3:

Every title of RESULTS was expressed unprofessionally! It should be a sentence with a verb!

Response:

As mentioned in the first response letter, we have changed the wording of each headline according to the suggestion of the reviewer.

Comment 10:

The manuscript is lack of a series of cellular results to validate the functions of this regulation!

Response:

As addressed in the first response letter, the focus of this article is the characterization of the promoter region of the Ct-SLCO1B3 gene. We agree with the reviewer that further studies regarding the possible function of this regulation may be important. However, such studies including e.g. CRISPR/Cas9 experiments to verify the influence of transcription factor binding sites on gene expression or protein expression are not in the scope of this manuscript.

For the complete Response Letter, please see the attachment.

Reviewer 3 Report

Accept in current form
